# RIEMANNIAN METRIC LEARNING VIA OPTIMAL TRANSPORT

**Christopher Scarvelis**
MIT CSAIL
scarv@mit.edu

**Justin Solomon**
MIT CSAIL
jsolomon@mit.edu

## ABSTRACT

We introduce an optimal transport-based model for learning a metric tensor from cross-sectional samples of evolving probability measures on a common Riemannian manifold. We neurally parametrize the metric as a spatially-varying matrix field and efficiently optimize our model's objective using a simple alternating scheme. Using this learned metric, we can nonlinearly interpolate between probability measures and compute geodesics on the manifold. We show that metrics learned using our method improve the quality of trajectory inference on scRNA and bird migration data at the cost of little additional cross-sectional data.

## 1 INTRODUCTION

In settings like single-cell RNA sequencing (scRNA-seq) (Tanay & Regev, 2017), we often encounter *pooled cross-sectional data*: Time-indexed samples $\left\{x_t^i\right\}_{i=1}^{N_t}$ from an evolving population $X_t$ with no correspondence between samples $x_s^i$ and $x_t^i$ at times $s \neq t$. Such data may arise when technical constraints impede the repeated observation of some population member. For example, as scRNA-seq is a destructive process, any given cell's gene expression profile can only be measured once before the cell is destroyed.

This data is often sampled sparsely in time, leading to interest in *trajectory inference*: Inferring the distribution of the population or the positions of individual particles between times $\{t_i\}_{i=1}^T$ at which samples are drawn. A fruitful approach has been to model the evolving population as a time-varying probability measure $P_t$ on $\mathbb{R}^D$ and to infer the distribution of the population between observed times by interpolating between subsequent pairs of measures $\left(P_{t_i}, P_{t_{i+1}}\right)$. Some existing approaches to this problem use dynamical optimal transport to interpolate between probability measures, which implicitly encodes a prior that particles travel along straight lines between observations. This prior is often implausible, especially when the evolving population is sampled sparsely in time.

One can straightforwardly extend optimal transport-based methods by allowing the user to specify a spatially-varying metric tensor to bias the inferred trajectories away from straight lines. This approach is theoretically well-founded and amounts to redefining a straight line by altering the manifold on which trajectories are inferred. Such a metric tensor, however, is typically unavailable in most real-world applications.

We resolve this problem by introducing an optimal transport-based model in which a metric tensor may be recovered from cross-sectional samples of evolving probability measures on a common manifold. We derive a tractable optimization problem using the theory of optimal transport on Riemannian manifolds, neurally parametrize its variables, and solve it using gradient-based optimization.

We demonstrate our algorithm's ability to recover a known metric tensor from cross-sectional samples on synthetic examples. We also show that our learned metric tensor improves the quality of trajectory inference on scRNA data and allows us to infer curved trajectories for individual birds from cross-sectional samples of a migrating population. Our method is both computationally-efficient, requiring little computational resources relative to the downstream trajectory inference task, and data-efficient, requiring little data per time point to recover a useful metric tensor.

## 2 RELATED WORK

**Measure interpolation.** An emerging literature considers the problem of smooth interpolation between probability measures. Using the theory of optimal transport, we may construct a *displacement interpolation* (McCann, 1997) between successive measures $(P_i, P_{i+1})$, yielding a sequence

of geodesics between pairs $(P_i, P_{i+1})$ in the space of probability measures equipped with the 2-Wasserstein distance $W_2$ (Villani, 2008). This generalizes piecewise-linear interpolation to probability measures. Schiebinger et al. (2019) use this method to infer the developmental trajectories of cells based on static measurements of their gene expression profiles. Recent works such as (De Bortoli et al., 2021) and (Vargas et al., 2021) provide numerical schemes for computing *Schrödinger bridges*; these are an entropically-regularized analog to the displacement interpolation between measures.

Chen et al. (2018b); Benamou et al. (2018) leverage the variational characterization of cubic splines as minimizers of mean-square acceleration over the set of interpolating curves and develop a generalization in the space of probability measures. Chewi et al. (2021) extend these works by providing computationally efficient algorithms for computing measure-valued splines.

Hug et al. (2015) modify the usual displacement interpolation between probability measures by introducing anisotropy to the domain on which the measures are defined. This change corresponds to imposing preferred directions for the local displacement of probability mass. Whereas Hug et al. hard-code the domain's anisotropy, our method allows us to *learn* this anisotropy from snapshots of a probability measure evolving in time. Zhang et al. (2022) apply similar techniques to unstructured animation problems.

Ding et al. (2020) propose a non-convex inverse problem for recovering the ground metric and interaction kernel in a class of mean-field games and supply a primal-dual algorithm for solving a grid discretization of the problem. Whereas their Eulerian approach inputs a grid discretization of observed densities and velocity fields, our approach is Lagrangian, operating directly on temporal observations of particle positions.

**Trajectory inference from population-level data.** In domains such as scRNA-seq, we study an evolving population from which it is impossible or prohibitively costly to collect longitudinal data. Instead, we observe distinct cross-sectional samples from the population at a collection of times and wish to infer the dynamics of the latent population from these observations. This problem is called *trajectory inference*.

Hashimoto et al. (2016) study the conditions under which it is possible to recover a potential function from population-level observations of a system evolving according to a Fokker-Planck equation. They provide an RNN-based algorithm for this learning task and investigate their model's ability to recover differentiation dynamics from scRNA-seq data sampled sparsely in time. A recent work of Bunne et al. (2022) presents a proximal analog to this approach, modeling population dynamics as a JKO flow with a learnable energy function and describing a numerical scheme for computing this flow using input-convex neural networks. This method addresses initial value problems where one is given an initial measure $\rho_0$ and seeks to predict future measures $\rho_t$ for $t > 0$. In contrast, our method is primarily applicable to boundary value problems, where we are given initial and final measures $\rho_0$ and $\rho_1$ and seek an interpolant $\rho_t$ for $0 < t < 1$.

Schiebinger et al. (2019) use optimal transport to infer future and prior gene expression profiles from a single observation in time of a cell's gene expression profile. Yang & Uhler (2019) propose a GAN-like solver for unbalanced optimal transport and investigate the effectiveness of their method for the inference of future gene expression states in zebrafish single-cell gene expression data.

As they learn transport maps between probability measures defined at discrete time points, none of the above OT-based methods is suitable for inferring continuous trajectories. Tong et al. (2020) show that a regularized continuous normalizing flow can provide an efficient approximation to displacement interpolations arising from dynamical optimal transport. These displacement interpolations often do not yield plausible paths through gene expression space, so the authors propose several application-specific regularizers to bias the inferred trajectories toward more realistic paths. Rather than relying on bespoke regularizers, our method supplies a natural approach for learning local directions of particle motion in a way that is amenable to integration with algorithms like that of Tong et al. (2020).

**Riemannian metric learning.** Lebanon (2002) develops a parametric method for learning a Riemannian metric from sparse high-dimensional data and uses this metric for nearest-neighbor classification. Hauberg et al. (2012) construct a metric tensor as a weighted average of a set of learned metric tensors and show how to compute geodesics and exponential and logarithmic maps on the resulting Riemannian manifold. Arvanitidis et al. (2016) learn a Riemannian metric that encourages geodesics to move towards region of high data density, define a Riemannian normal distribution with respect to

this metric, and show how to learn the parameters of this model via maximum likelihood estimation. Whereas these methods learn a Riemannian metric from *static* data, our model learns a metric from snapshots of probability distributions evolving in time on a common manifold.

## 3 METHOD

We now describe our method for learning a Riemannian metric from cross-sectional samples of populations evolving in time on a common manifold. We learn a metric that minimizes the average 1-Wasserstein distance on the manifold between pairs of subsequent time samples from each population. We derive a dual formulation of our problem, parametrize its variables by neural networks, and solve for the dual variables and the metric via alternating optimization.

### 3.1 MODEL

Suppose we have $K$ populations evolving according to unknown continuous dynamics over a common Riemannian manifold $\mathcal{M} = \left( \mathbb{R}^D, g \right)$ with unknown metric $g$. The metric $g$ is defined at any $x \in \mathbb{R}^D$ by the inner product $\langle u, v \rangle_x = u^T A(x) v$ for $A(x) \succ 0$. We model each population as a compactly-supported probability distribution $P^k$ with density $\rho^k$ on $\mathbb{R}^D$ being pushed through an unobserved velocity field $v^k(x)$. We will *learn* the metric tensor $A(x)$ from temporal snapshots of the populations $P^k$ during their evolution on the manifold.

Depending on the nature of our data, we may have the ability to repeatedly sample from $P^k$ at a pair of initial and final times $t = 0$ and $t = T_k$ or have a fixed set of samples $\{x_i^{k,0}\}_{i=1}^{S^k}$ and $\{x_i^{k,T_k}\}_{i=1}^{S^k}$ drawn from the populations $P^k$ at their respective times. For convenience, we denote both the density $\rho^k$ and the empirical distributions over samples from $P^k$ at times $t \in \{0, T_k\}$ by $\rho_0^k$ and $\rho_1^k$ respectively. As we only observe the initial and final spatial distributions $\rho_0^k$ and $\rho_1^k$ of the populations and do not observe their dynamics, we assume that probability mass travels from initial to final positions along $A$-geodesics. Geodesics are paths that minimize the *action* (or average kinetic energy) of a particle traveling between points $x, y \in \mathcal{M}$; this least-action interpretation of a geodesic makes it a natural prior on paths in the absence of further information. We learn a field of positive definite matrices $A(x)$ that minimizes the average $A$-geodesic distance between the initial and final positions of each unit of probability mass from each $P^k$.

Formally, let $r^k$ be the map sending a point $x \in \mathcal{M}$ to its final position $r^k(x)$ after flowing through the latent velocity field $v^k$ from time $t = 0$ to $t = T_k$. If we had access to such a solution map, we would ideally solve the following problem:

$$\inf_{A:\mathbb{R}^D \to S_{++}^D} \frac{1}{K} \sum_{k=1}^K \int_{\mathcal{M}} d_A\left(x, r^k(x)\right) d\rho_0^k(x) + \lambda R\left(A\right), \tag{1}$$

where $R(A)$ is a regularizer that excludes the trivial solution $A \equiv 0$. Problem (1) optimizes for a non-trivial metric $A(x)$ that minimizes the average $A$-geodesic distance $d_A\left(x, r^k(x)\right)$ traveled by particles $x$ in the population $\rho_0^k$ at time $t = 0$ to their final positions $r^k(x)$ at time $t = T_k$.

Since we do not know the velocity fields $v^k$ that encode the particle dynamics, however, we also do not know the maps $r^k$ in (1) that specify the correspondence between particle positions $x$ at $t = 0$ and positions $r^k(x)$ at final times $T_k$. Furthermore, as noted in Section 1, we often encounter data for which it is impossible to observe contiguous particle trajectories: A particle that we observe at $t = 0$ may not be in the sample at $t = T_k$. This issue is unavoidable for destructive measurement processes such as scRNA sequencing; in this setting, a cell whose scRNA profile is observed at $t = 0$ would be destroyed at this time and hence unobservable at a future time $t = T_k$. To accommodate these limitations, we replace the true matchings of initial and final positions $r^k$ with the *Monge map* $s^k$, defined as the solution to the following problem:

$$W_{1,A}(\rho_0^k, \rho_1^k) = \inf_{s^k : \rho_1^k = s_\#^k \rho_0^k} \int_{\mathcal{M}} d_A\left(x, s^k(x)\right) d\rho_0^k(x). \tag{2}$$

Here $s^k$ is a *pushforward* of $\rho_0^k$ onto $\rho_1^k$; we write $\rho_1^k = s_\#^k \rho_0^k$ to denote this relationship. This map matches units of mass from the initial and final distributions $\rho_0^k, \rho_1^k$ to minimize their average $A$-geodesic distance. Substituting solutions to (2) for the maps $r^k$ in our idealized objective (1), we

obtain the following lower bound on (1):

$$\inf_{A:\mathbb{R}^D \to S_{++}^D} \underbrace{\inf_{s^k:\rho_1^k = s_\#^k \rho_0^k} \frac{1}{K} \sum_{k=1}^{K} \int_{\mathcal{M}} d_A\left(x, s^k(x)\right) d\rho_0^k(x)}_{= \frac{1}{K} \sum_{k=1}^{K} W_{1,A}(\rho_0^k, \rho_1^k)} + \lambda R\left(A\right). \tag{3}$$

Problem (3) is challenging as written: It requires the ability to compute and differentiate geodesic distances $d_A$ with respect to an arbitrary metric. However, the inner optimization problem over maps $s^k$ is a collection of decoupled Monge problems (2) whose optimal value is the average 1-Wasserstein distance between $(\rho_0^k, \rho_1^k)$ pairs on the manifold with metric $A(x)$. As shown in Appendix B, these problems can be expressed in a dual form (10) which is amenable to gradient-based optimization.

Replacing the inner Monge problems with their dual formulations (10), we may equivalently write Problem (3) as a minimax problem:

$$\inf_{A:\mathbb{R}^D \to S_{++}^D} \sup_{\substack{\phi^k:\mathcal{M} \to \mathbb{R} \\ \|\nabla\phi^k(x)\|_{A^{-1}(x)} \leq 1}} \frac{1}{K} \sum_{k=1}^{K} \left( \int_{\mathcal{M}} \phi^k(x) d\rho_0^k(x) - \int_{\mathcal{M}} \phi^k(x) d\rho_1^k(x) \right) + \lambda R\left(A\right). \tag{4}$$

In Problem (4), we learn a metric $A(x)$ that minimizes the average 1-Wasserstein distance on the manifold between pairs of subsequent time samples from each population. (4) requires neither the computation of geodesic distances on $\mathcal{M}$ nor the solution of an assignment problem. As such, it is substantially more tractable than the initial objective (3). We provide the details of our implementation of (4) in Section 3.2.

## 3.2 IMPLEMENTATION

**Enforcing the Lipschitz constraint.** Problem (4) includes global constraints of form $\|\nabla\phi^k\|_{A^{-1}} \leq 1$. These constraints are the Riemannian analog to the Lipschitz constraint in the dual formulation of the 1-Wasserstein distance on $\mathbb{R}^D$. Constraints of this type are challenging to enforce in gradient-based optimization, and the Wasserstein GAN literature has explored approximations (Arjovsky et al., 2017; Gulrajani et al., 2017; Miyato et al., 2018). We follow the standard technique introduced by Gulrajani et al. (2017) and replace the global constraints $\|\nabla\phi^k\|_{A^{-1}} \leq 1$ with soft penalties of the following form:

$$\mathbb{E}_{\substack{x_0 \sim \rho_0^k \\ x_1 \sim \rho_1^k \\ t \sim U(0,1)}} \left[ \text{SoftPlus} \left( \|\nabla\phi^k\left(\sigma_{x_0}^{x_1}(t)\right)\|_{A^{-1}\left(\sigma_{x_0}^{x_1}(t)\right)}^2 - 1 \right) \right], \tag{5}$$

where $\sigma_{x_0}^{x_1}(t) := (1-t)x_0 + tx_1$ is a line segment between $x_0$ and $x_1$ parametrized by $t \in [0,1]$. Intuitively, (5) penalizes violation of the Lipschitz constraint $\|\nabla\phi^k\|_{A^{-1}} \leq 1$ along line segments connecting randomly-paired points in $X_0^k$ and $X_1^k$. Gulrajani et al. (2017, Prop. 1) justify this choice via a standard result in $W_1$ theory showing that the constraint $\|\nabla\phi^k\| \leq 1$ binds on line segments connecting pairs of points that are matched by the Monge map $s^k$. Korotin et al. (2022) show that this method results in accurate approximations to the directions of the gradients $\nabla\bar\phi^k$ of the true Kantorovich potentials. This observation is sufficient for our purposes; as noted below, our optimization scheme encourages the low-energy eigenvectors of $A(x)$ to be well-aligned with the solutions $\nabla\phi^k$ to the inner problem in (4).

**Choice of regularizer $R(A)$.** Without a regularizer $R(A)$, the objective in (4) can be driven to 0 by choosing $A(x) \equiv \alpha I$ for arbitrarily small $\alpha > 0$, thereby making all pairs of measures arbitrarily close. This trivial solution would incorporate no useful information from the observed samples from $\rho_0^k$ and $\rho_1^k$, as it is simply a rescaling of the standard Euclidean metric.

We opt for the following regularizer in our method:

$$R(A) = \frac{1}{K} \sum_{k=1}^{K} \mathbb{E}_{\substack{x_0 \sim \rho_0^k \\ x_1 \sim \rho_1^k \\ t \sim U(0,1)}} \left[ \|A^{-1}\left(\sigma_{x_0}^{x_1}(t)\right)\|_F^2 \right], \tag{6}$$

where $\sigma_{x_0}^{x_1}(t)$ is as in the previous paragraph and $\|\cdot\|_F^2$ denotes the squared Frobenius norm.

Regularizer (6) penalizes $\|A^{-1}\|_F^2$ at points drawn using a sampling scheme analogous to that in (5). This is a natural way to exclude trivial solutions of form $A(x) \equiv \alpha I$ for arbitrarily small $\alpha > 0$, as $\|A^{-1}\|_F^2$ is large for such metrics. We investigate the impact of the regularization coefficient $\lambda$ on the learned metric in Appendix E. In Appendix F, we demonstrate the effect of removing the optimal transport term from (4), which is equivalent to setting $\lambda = +\infty$.

**Optimization scheme.** We parametrize the scalar potentials $\phi^k$ by neural networks to enable the use of gradient-based optimization to solve Problem (4). As $A$ appears in Equations (4), (5), and (6) only via its inverse $A^{-1}$, we directly parametrize the matrix field $A^{-1}$ by a neural network; where we require the evaluation of $A(x)$ in downstream applications, we evaluate and then invert $A^{-1}(x)$. We enforce the positive definiteness of $A^{-1}$ (and hence the positive definiteness of $A$) by parametrizing it as $A^{-1}(x) = Q(x)^T Q(x) + \eta I$ for a matrix-valued function $Q(x) : \mathcal{M} \to \mathbb{R}^{D \times D}$ and $\eta > 0$.

After parametrizing our problem variables with neural networks, we optimize the objective in (4) via alternation. In the first phase of our scheme, we solve the inner problem by holding $A$ fixed (initializing it as $A(x) = A^{-1}(x) \equiv I$) and solving for the optimal $\phi^k$. This step decouples over the potentials $\phi^k$, and each of the resulting problems is an instance of the dual problem for the 1-Wasserstein distance on $\mathcal{M}$. We approximate the integrals in (4) as sample means $\frac{1}{N}\sum_{i=1}^{N}\phi^k(x_i)$, where the $\{x_i\}$ are samples from the distributions $\rho_0^k$ and $\rho_1^k$. We likewise approximate the Lipschitz penalty (5) and the regularizer (6) as sample means over draws from $\rho_0^k$ and $\rho_1^k$ and over $t$ drawn uniformly from $[0, 1]$. In the second phase, we solve the outer problem by fixing the optimal $\phi^k$ from the previous step and solving for the optimal matrix field $A(x)$. We optimize both problems using AdamW (Loshchilov & Hutter, 2019). Few alternations are needed in practice to obtain high-quality results.

The results in Appendix B show that given a fixed metric defined by $A(x)$, the optimal $\nabla\phi^k$ from the first phase of our scheme point along $A$-geodesics joining pairs of points $(x, s^k(x))$ where $x \sim \rho_0^k$ and $s^k$ solves (2). At initialization, $A \equiv I$, and these geodesics are line segments in $\mathbb{R}^D$ connecting the matched points.

Given fixed Kantorovich potentials $\phi^k$ from the first phase, the second phase of our scheme solves for a matrix field $A(x)$ that minimizes the regularizer $R(A) = \|A^{-1}\|_F^2$ while also making $\|\nabla\phi^k\|_{A^{-1}}$ large. Intuitively, this encourages the unit eigenvector $u_1(x)$ corresponding to the minimal eigenvalue $\lambda_1(x)$ of $A(x)$ to be aligned with $\nabla\phi^k(x)$ wherever the constraint is enforced.

## 4 EXPERIMENTS

In this section, we first use synthetic data to demonstrate that our algorithm successfully recovers the correct eigenspaces of a known metric $A(x)$ from cross-sectional samples satisfying our model. We then use our method to learn a metric from cross-sectional scRNA data and show that this metric improves the accuracy of trajectory inference for scRNA data that is sampled sparsely in time. We finally show that by learning a metric from time-stamped bird sightings, we can infer curved migratory trajectories for individual birds given the initial and final points of their trajectories. Details for all experiments are provided in Appendix D.

### 4.1 METRIC RECOVERY

We first show that our method recovers the correct eigenspaces of a known metric $A(x)$ from cross-sectional samples $X_0^k$ and $X_1^k$ satisfying the model in Section 3.1. We fix initial positions $x_n(0)$ and final positions $x_n(1)$ for a set of $N$ particles $x_n \in X$ and also fix a spatially-varying metric $A(x)$. We compute $A$-geodesics between each $(x_n(0), x_n(1))$ pair and define populations $X^{t_i} = \{x_n(t_i) : n = 1, ..., N\}$ for times $t_i \in [0, 1]$, $i = 0, ..., T$.

We use our method to learn the latent metric $A(x)$ from the pairs of samples $(X^{t_i}, X^{t_{i+1}})$. For each example, we plot the eigenvectors of the true metric $A(x)$ and those of the learned metric $\hat{A}(x)$ on a $P \times P$ grid (first row, Figure 1). We also plot the eigenvectors of $\hat{A}(x)$ along with the log-condition number $\log(\lambda_2(x)/\lambda_1(x))$ of $\hat{A}(x)$ (second row, Figure 1); this value is large when the learned metric is highly anisotropic. We finally report a measure of the alignment of the eigenspaces of the true metric $A$ and its learned counterpart: $\ell(A, \hat{A}) = \frac{1}{D|\mathcal{X}|}\sum_{x \in \mathcal{X}}\sum_{d=1}^{D}|\langle u_d(x), \hat{u}_d(x)\rangle|$. Here $u_d(x)$ is the unit eigenvector with eigenvalue $\lambda_d$ of $A(x)$, $\hat{u}_d(x)$ is the corresponding eigenvector for $\hat{A}(x)$,

and $\mathcal{X}$ is a set of grid points at which we plot the eigenvectors of $A$ and $\hat{A}$. $\ell(A, \hat{A}) = 1$ when each eigenvector of $A(x)$ is parallel to the corresponding eigenvector of $\hat{A}(x)$ at all grid points $x \in \mathcal{X}$.

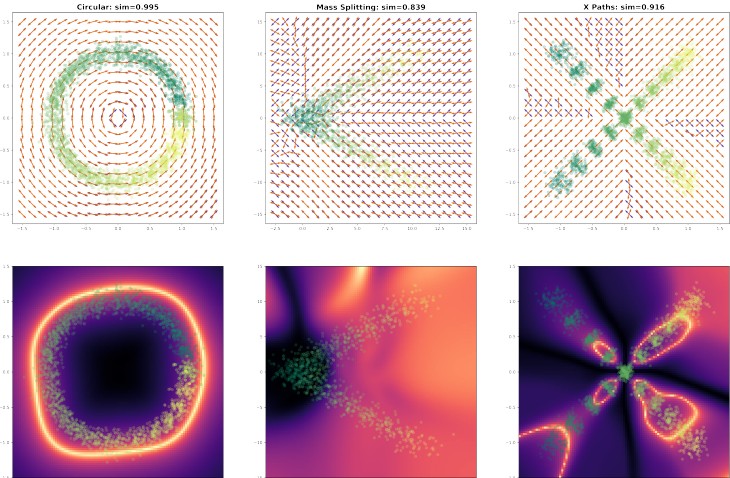

Figure 1: Row 1: Eigenvectors of true metric $A(x)$ (purple) and learned metric $\hat{A}(x)$ (orange). Row 2: Log-condition number of learned metric $\hat{A}(x)$ - yellow indicates highly anisotropic $\hat{A}(x)$. The points are time samples from which $\hat{A}(x)$ is recovered. Teal points indicate earlier times and yellow points indicate later times, so the pairs of temporal samples $(X^{t_i}, X^{t_{i+1}})$ follow the color gradient. Our method accurately recovers the eigenvectors of the true metric, and the learned metric is highly anisotropic in regions that overlap the observed data.

Our method accurately recovers the eigenspaces of the "Circular" ($\ell(A, \hat{A}) = 0.995$) and "X Paths" ($\ell(A, \hat{A}) = 0.916$) metrics. It achieves lower accuracy with the "Mass Splitting" metric ($\ell(A, \hat{A}) = 0.839$), struggling to capture the discontinuity in its eigenvectors at the $x$-axis and exhibiting numerical instability to the left of the $y$-axis, where the training trajectories diverge. Note, however, that the alignment score $\ell(A, \hat{A})$ is in part measured at points that do not lie on the trajectory of the training data. We would not expect our method to accurately recover the metric in these regions; the fact that it largely does so for the "Circular" and "X Paths" examples reflects desirable smoothness properties of the neural parametrization of the Kantorovich potentials and the metric tensor.

Row 2 of Figure 1 shows that our learned metric is highly anisotropic in regions that overlap with the training data. In the "Mass Splitting" and "X Paths" examples, $\hat{A}(x)$ also has small condition number near the origin where the two paths diverge and cross, respectively; this behavior reflects expected uncertainty in low-energy directions of motion in this region.

## 4.2 POPULATION-LEVEL TRAJECTORY INFERENCE WITH A LEARNED METRIC

scRNA sequencing (scRNA-seq) allows biologists to study the set of mRNA molecules (the "transcriptome") of individual cells at high resolution (Haque et al., 2017). As scRNA-seq is a destructive process, any individual cell's RNA can be sequenced only once, impeding the use of this technology to study the change in a cell's transcriptome over time. This has led to interest in methods that use population-level scRNA-seq data to *infer* the temporal evolution of an individual cell's scRNA-seq profile.

Optimal transport-based techniques are well-established tools for solving these trajectory inference problems. Schiebinger et al. (2019) and Tong et al. (2020) both use optimal transport to infer cellular development trajectories but assume a Euclidean metric on gene expression space. In this section, we use our method to relax this strong assumption by learning a metric for the ground space from scRNA data. We incorporate $\hat{A}(x)$ into a downstream trajectory inference task and show that this strategy yields more accurate trajectories than a baseline without a learned metric and a baseline using the Euclidean metric.

**scRNA data.** We perform trajectory inference experiments with the scRNA data drawn from Schiebinger et al. (2019). This data consists of force-directed layout embedding coordinates of gene expression data collected over 18 days of reprogramming (39 time points total). We construct populations $X^{t_i}$ for $i = 1, \ldots, 39$ by drawing 500 samples per time point in the original data; this sampling uses 8.25% of the available data on average. We follow the same procedure as in Section 4.1 to learn a metric tensor $\hat{A}(x)$ from subsequent pairs of samples $(X^{t_i}, X^{t_{i+1}})$. Learning the tensor takes 16 minutes on a single V100 GPU.

For the downstream trajectory inference task, we keep one out of every $k$ time points in the original data for $k = 2, ..., 19$ to obtain a new collection of time points $\bar{t}_\ell$ with $\ell \in \{nk : n \in \mathbb{N}, n < \frac{39}{k}\}$. We then perform trajectory inference between subsequent retained time points $(\bar{t}_\ell, \bar{t}_{\ell+1})$ (using all of the available data for these time points) by optimizing the following objective:

$$\min_\theta S_\epsilon \left( X^{\bar{t}_{\ell+1}}, \phi_{t=1}(v_\theta)[X^{\bar{t}_\ell}] \right) + \lambda \sum_{x \in X^{\bar{t}_\ell}} \sum_{j=1}^{m} ||v_{t,\theta}(x(t_j))||^2_{\hat{A}(x(t_j))}. \tag{7}$$

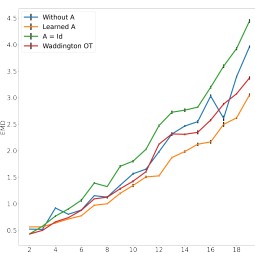

Here $S_\epsilon$ is the Sinkhorn divergence (Feydy et al., 2019) between the target data $X^{\bar{t}_{\ell+1}}$ and the advected samples $\phi_{t=1}(v_\theta)[X^{\bar{t}_\ell}]$; $\phi_{t=1}(v_\theta)$ denotes the solution map resulting from advecting a particle through a neurally-parametrized time-varying velocity field $v_{t,\theta}$ for one unit of time; and $t_j \in [0, 1]$. The Sinkhorn divergence used here is $S_\epsilon(\alpha, \beta) = W_\epsilon(\alpha, \beta) - \frac{1}{2} W_\epsilon(\alpha, \alpha) - \frac{1}{2} W_\epsilon(\beta, \beta)$; it is a debiased variant of the entropically-regularized 2-Wasserstein distance $W_\epsilon$.

This is similar to the method of Tong et al. (2020), where we replace the log-likelihood fitting loss with the Sinkhorn divergence; we found that the Sinkhorn divergence led to stabler training than a log-likelihood fitting loss. We use GeomLoss (Feydy et al., 2019) to compute the Sinkhorn divergence between the target and advected samples efficiently.

Figure 2: Mean EMD between left-out samples and corresponding advected samples versus $k$, with our learned metric (orange), the Euclidean metric $A(x) \equiv I$ (green), no regularizer (blue), and interpolants obtained using Schiebinger et al. (2019)'s method (red).

We follow Tong et al. (2020) and assess the quality of inferred trajectories by measuring the $W_1$ distance (EMD) between left-out time points in the ground truth data and advected samples at corresponding times in the inferred trajectories. These distances are of the form $W_1 \left( X^{t_i}, \phi_{t=\hat{t}_i}(v_\theta)[X^{\bar{t}_\ell}] \right)$, so we compare ground truth samples at each left-out time $t_i = nk + h$ ($h = 1, ..., k - 1$) with samples from $X^{\bar{t}_\ell}$ advected through $v_\theta$ for time $\hat{t}_i = \frac{h}{k}$. For each value of $k$, we record the average EMD between left out samples and their inferred counterparts and compare our method to a baseline where $\lambda = 0$ in (7) ("Without $A$"

in Figure 2) and a Euclidean baseline where $A(x) \equiv I$ ("$A = $ Id" in Figure 2). Our "Without $A$" and "$A = $ Id" baselines are comparable to the "Base" and "Base + E" models, respectively, from Tong et al. (2020). Both our baseline models and their models learn a velocity field $v_{t,\theta}$ that pushes samples at some time $\bar{t}_\ell$ onto samples at a future time $\bar{t}_{\ell+1}$ and use the path followed by these samples as they flow through $v_{t,\theta}$ as an inferred trajectory. Our "Without $A$" baseline and their "Base" model do not regularize $v_{t,\theta}$, whereas our "$A = $ Id" baseline and their "Base + E" model regularize $v_{t,\theta}$ by penalizing its squared norm, which encourages samples to flow along straight paths.

We also compare against interpolants obtained via the method of Schiebinger et al. (2019). As their Waddington OT method solves a static optimal transport problem and pushes cells through transport maps between fixed time steps, a direct comparison against our dynamical method is not possible. However, we attempt to replicate their validation by geodesic interpolation as closely as possible. We compute static optimal transport couplings between data at subsequent retained time points $(\bar{t}_\ell, \bar{t}_{\ell+1})$, linearly interpolate between coupled data points, and measure the $W_1$ distance (EMD) between left-out time points in the ground truth data and advected samples at corresponding times in the interpolated trajectories.

Figure 2 shows that our learned metric improves on the "Without $A$" baseline (analogous to the "Base" model of Tong et al. (2020)) and the approach based on Schiebinger et al. (2019) for nearly all values of $k$, whereas the Euclidean baseline $A(x) \equiv I$ (analogous to the "Base + E" model of Tong et al. (2020)) generally results in less accurate trajectories than the non-regularized baseline. As expected, the gap between our results and both baselines increases for large values of $k$; for example,

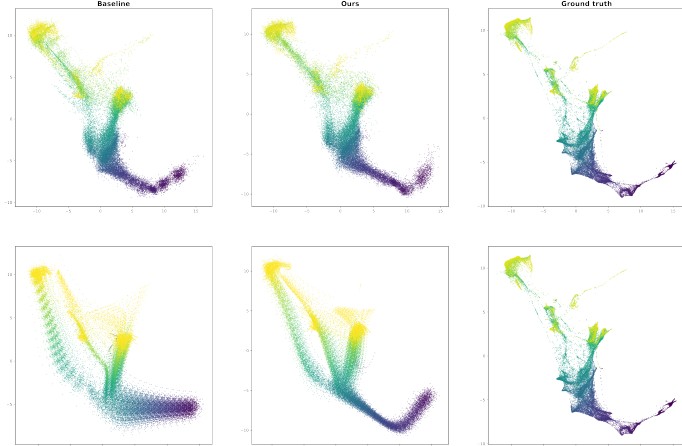

Figure 3: Comparison of inferred trajectories for $k = 3$ (first row) and $k = 15$ (second row). Trajectories inferred using our learned metric tensor (second column) more closely follow the manifold structure of the ground truth data than the non-regularized baseline trajectories (first column), where particles follow nearly straight-line paths between observed time points.

using our learned metric results in a 19.7% mean reduction in EMD between left-out samples and the corresponding advected samples for $k \geq 10$. This observation indicates that including a learned metric has a larger impact on the inferred trajectories when the ground truth data is sampled sparsely in time.

Figure 3 compares the trajectories inferred using our method to the non-regularized baseline and ground truth trajectories for $k = 3$, where the time sampling is sufficiently dense that the baseline performs well, and $k = 15$, where our learned metric tensor substantially improves the quality of the inferred trajectories. Whereas the non-regularized baseline simply advects particles from the base measures to the corresponding targets along nearly-straight paths, the learned metric biases the trajectory to follow the highly-curved paths taken by the ground truth data.

These results suggest that in settings where data collection is expensive and samples collected at a small subset of times are of primary interest, our method enables the plausible inference of particle positions at intermediate time points at the cost of little additional data and computation.

### 4.3 INDIVIDUAL-LEVEL TRAJECTORY INFERENCE WITH A LEARNED METRIC

In this section, we learn a metric $\hat{A}(x)$ from time-stamped sightings of snow geese during their spring migration. We then compute an $\hat{A}$-geodesic between the initial and final point of GPS-tagged snow geese and show that this provides a reasonable coarse approximation to the geese's ground truth trajectories.

**Migratory paths as geodesics on a latent manifold.** Many migratory bird species return to their summer breeding sites and wintering grounds with high spatial fidelity, sometimes returning to within 500 meters of their usual breeding location (Mowbray et al., 2020). Given this behavior, a boundary value problem that fixes the endpoints of the birds' migratory trajectories is a reasonable model for bird migration. Once the endpoints are fixed, we must specify an objective that birds plausibly optimize to determine their trajectories. Absent further information, minimizing total kinetic energy over their trajectory is a reasonable choice. However, measuring kinetic energy with respect to the Euclidean metric leads to straight-line paths, which typically do not agree with observed migration trajectories. We use our method to learn a metric from untagged snow goose sightings that agrees with the data for this species. This metric summarizes the factors that birds may use to modify their migratory paths locally, such as local weather conditions, food availability, and predatory pressures.

**Snow goose data.** The training data for this experiment consists of time-stamped sightings of *untagged* snow geese (*Anser caerulescens*) across the U.S. and Canada during their spring migration. We treat the sightings at time $t$ as samples from a time-indexed spatial distribution of birds $\rho^t$ and train our metric on subsequent pairs of bird distributions $(\rho^t, \rho^{t+1})$. This implies that *our method does not have access to any complete goose trajectories when learning the metric*. We do not expect untagged goose sightings to contain enough information to predict high-frequency detail in migratory trajectories, but this data is far cheaper to obtain than complete migratory trajectories, which are typically recorded via GPS trackers attached to individual geese. The widespread availability and

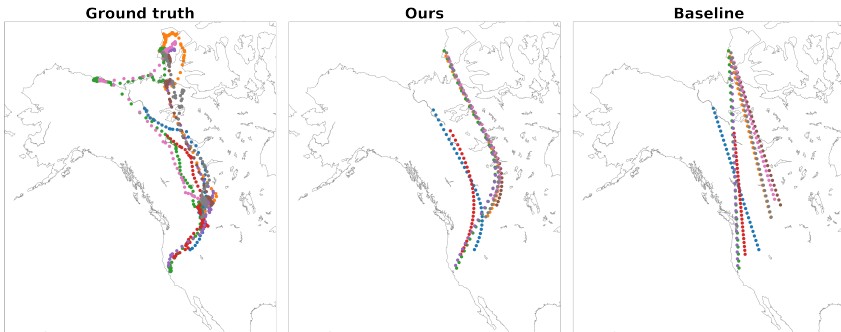

Figure 4: By using a metric $\hat{A}(x)$ learned from time-stamped bird sightings, we obtain inferred trajectories (center) that capture the curved structure of the ground truth migratory paths (left). Our method results in a 26.9% reduction in mean DTW distance between the inferred and ground truth trajectories relative to the Euclidean baseline (right).

low cost of obtaining untagged bird sighting data motivates the use of our method for bird trajectory inference

The training data is drawn from the eBird basic dataset (Sullivan et al., 2009), current as of February 2022. We bin the sightings by month of observation and use our algorithm to learn a metric tensor $\hat{A}(x)$ from populations $X^i$ consisting of the spatial coordinates of snow goose sightings in month $i$ for $i = 0, ..., 5$ (i.e. January to June). This training data is depicted in Figure 5 in Appendix D.3.

We use this learned metric tensor to compute an $\hat{A}$-geodesic between the initial and final observation of several snow geese along their migratory paths. This data is drawn from the Banks Island Snow Goose study as hosted on Movebank (Kays et al., 2021). We estimate an $\hat{A}$-geodesic between initial and final points on each path by learning a time-varying velocity field $v_{t,\theta}$ optimizing the following objective:

$$\min_{\theta} \|x_1 - x(1)\|_2 + \mu \sum_{j=1}^{m} ||v_{t,\theta}(x(t_j))||^2_{\hat{A}(x(t_j))}. \tag{8}$$

As in Section 4.2, the velocity field $v_{t,\theta}$ is neurally parametrized, $0 = t_0 \leq \cdots t_j \leq \cdots t_m = 1$, and we optimize (8) using AdamW. We set the initial condition $x(0) = x_0$ such that $x_0$ is the goose's initial position on its migratory trajectory and use a Euclidean norm penalty to force its final position $x(1)$ along the inferred trajectory to match the true final position $x_1$.

Figure 4 compares the $\hat{A}$-geodesics obtained using the method described above (center plot) to the ground truth goose trajectories (left plot). While population-level data (in the form of untagged snow goose sightings; see Figure 5 in Appendix D.3) does not provide sufficient information to reconstruct the migratory paths of individual geese perfectly, the inferred trajectories accurately capture the curved structure of the ground truth trajectories.

We evaluate our method's performance by computing the dynamic time warping (DTW) distance (Berndt & Clifford, 1994) between each inferred goose trajectory and the corresponding ground truth trajectory. We also generate straight-line paths (i.e. Euclidean geodesics) between the initial and final points of each ground truth goose trajectory and compute DTW distances between these paths and the ground truth. *Our method results in a 26.9% reduction in mean DTW distance* between the inferred and ground truth trajectories relative to the Euclidean baseline.

## 5 DISCUSSION AND CONCLUSION

We have introduced an optimal transport-based method for learning a Riemannian metric from cross-sectional samples of populations evolving over time on a latent Riemannian manifold. Our method accurately recovers metrics from cross-sections of populations moving along geodesics on a manifold, improves the quality of trajectory inference on sparsely-sampled scRNA data at low data and compute cost, and allows us to approximate individual trajectories of migrating birds using information from untagged sightings.

One key limitation of our work is that it learns a Riemannian metric on $\mathbb{R}^D$, whereas large swaths of data naturally lie on graphs. Future work might consider extending this algorithm learn from data defined on graphs; a recent work of Heitz et al. (2021) presents an optimal transport-based direction towards solving this problem. Future work may also extend our methods to learn a metric from temporal snapshots of Schrödinger bridges.

## 6 ACKNOWLEDGEMENTS

The MIT Geometric Data Processing group acknowledges the generous support of Army Research Office grants W911NF2010168 and W911NF2110293, of Air Force Office of Scientific Research award FA9550-19-1-031, of National Science Foundation grants IIS-1838071 and CHS-1955697, from the CSAIL Systems that Learn program, from the MIT–IBM Watson AI Laboratory, from the Toyota–CSAIL Joint Research Center, from a gift from Adobe Systems, and from a Google Research Scholar award.

CS acknowledges the support of the Natural Sciences and Engineering Research Council of Canada (NSERC), funding reference number CGSD3-557558-2021, and from the 2022 Siebel Scholars award.

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

## A    Background on Riemannian geometry

A *Riemannian manifold* $\mathcal{M} = (M, g)$ is a differentiable manifold $M$ equipped with a *Riemannian metric* $g$. Throughout this paper, we take $M = \mathbb{R}^D$ and focus our attention on the metric $g$. A Riemannian metric assigns an inner product $\langle \cdot, \cdot \rangle_p$ to the tangent space $T_p M$ of each $p \in M$ in a way that varies smoothly with $p$. In the case where $M = \mathbb{R}^D$, we may simply identify all tangent spaces $T_p M$ with $\mathbb{R}^D$; here a metric $g$ amounts to a spatially-varying inner product on $\mathbb{R}^D$. Since any inner product on $\mathbb{R}^D$ may be computed as $\langle u, v \rangle_A = u^T A v$ for some $A \succ 0$, a Riemannian metric $g$ on $M = \mathbb{R}^D$ is specified by a smooth field of positive definite matrices $A(x) : \mathbb{R}^D \to S_{++}^D$ (we use $S_{++}^D$ to denote the set of positive definite $D \times D$ matrices). We use $\langle u, v \rangle_{A(x)} = u^T A(x) v$ to denote the metric at a point $x \in \mathbb{R}^D$.

An inner product $\langle \cdot, \cdot \rangle_{A(x)}$ induces a norm $\|v\|_{A(x)} := \sqrt{\langle v, v \rangle_{A(x)}}$. Given this norm, we define the *length* of a continuously differentiable curve $\gamma : [0, 1] \to \mathcal{M}$ to be $\ell(\gamma) = \int_0^1 \|\dot{\gamma}(t)\|_{A(\gamma(t))} \, dt$. The *A-geodesic distance* $d_A(x, y)$ between two points $x, y \in \mathcal{M}$ is then the infimum of the length $\ell(\gamma)$ of continuously differentiable curves $\gamma$ such that $\gamma(0) = x$ and $\gamma(1) = y$.

## B    $W_1$ on Riemannian manifolds

Feldman & McCann (2002) study *Monge's transport problem* on a Riemannian manifold $\mathcal{M} = (M, g)$; we focus on the case $M = \mathbb{R}^D$ with $g$ specified by a field of matrices $A(x) \succ 0$. Given two

compactly-supported densities $\rho_0, \rho_1$ on $\mathcal{M}$, Monge's problem seeks a pushforward $s$ of $\rho_0$ onto $\rho_1$ solving the following problem:

$$W_{1,A}(\rho_0, \rho_1) = \inf_{s:\rho_1 = s_\# \rho_0} \int_{\mathcal{M}} d_A\left(x, s(x)\right) d\rho_0(x). \tag{9}$$

Feldman & McCann (2002) show that under mild technical conditions, (9) has a possibly non-unique solution $s$. Intuitively, this map is a pushforward of $\rho_0$ onto $\rho_1$ minimizing the average geodesic distance between matched units of probability mass. The optimal value of (9) is the 1-Wasserstein distance between $\rho_0$ and $\rho_1$, which we denote by $W_{1,A}(\rho_0, \rho_1)$.

If $\phi$ is continuously differentiable on $\mathcal{M}$, then it is 1-Lipschitz on $\mathcal{M}$ if and only if $\|\nabla\phi(x)\|_{A^{-1}(x)} \leq 1$ for all $x \in \mathcal{M}$. Here, we use $\nabla\phi(x)$ to denote the vector of partial derivatives of $\phi$ at $x$. Armed with this local characterization of Lipschitz continuity, we may define the following dual problem to (9):

$$\sup_{\substack{\phi:\mathcal{M}\to\mathbb{R} \\ \|\nabla\phi(x)\|_{A^{-1}(x)}\leq 1}} \left(\int_{\mathcal{M}} \phi(x)d\rho_0(x) - \int_{\mathcal{M}} \phi(x)d\rho_1(x)\right). \tag{10}$$

If the minimum in (10) is attained by some *Kantorovich potential* $\phi$, the optimal values of (9) and (10) coincide and the Lipschitz bound for $\phi$ is tight at pairs of points $(x, y)$ arising from a Monge map $s$ solving (9) (Feldman & McCann, 2002, Lemma 4).

Feldman & McCann (2002, Lemma 10) also show that given a geodesic $\gamma_x : [0, 1] \to \mathcal{M}$ between a pair of matched points $(x, s(x))$ and $t \in (0, 1)$, we have $\nabla\phi\left(\gamma_x(t)\right) = -\frac{\dot{\gamma}_x(t)}{\|\dot{\gamma}_x(t)\|_{A(\gamma_x(t))}}$. Intuitively, $\nabla\phi$ points along geodesics $\gamma_x$ joining pairs of points in the support of $\rho_0$ and $\rho_1$ that are matched by the Monge map $s$.

## C  BACKGROUND ON DYNAMICAL OPTIMAL TRANSPORT

Let $\alpha$ and $\beta$ be probability measures defined on $\Omega \subseteq \mathbb{R}^D$. The Benamou-Brenier formulation of the 2-Wasserstein distance $W_2(\alpha, \beta)$ defines it as the solution to a fluid-dynamical problem (Benamou & Brenier, 2000)

$$W_2^2(\alpha, \beta) = \min_{\rho_t, v_t} \int_0^1 \int_\Omega \rho_t(x)\|v_t(x)\|_2^2 \, dx \, dt \tag{11}$$

subject to the constraints that $\rho_0 = \alpha$, $\rho_1 = \beta$ and the continuity equation $\frac{\partial \rho_t}{\partial t} + \nabla \cdot (\rho_t v_t) = 0$.

Solving this problem yields a time-varying velocity field $v_t(x)$ that transports $\alpha$'s mass to $\beta$'s along a curve $\rho_t$ of probability measures. The kinetic energy in the integrand of (11) encourages probability mass to travel in straight lines. This assumption is occasionally undesirable, but it is straightforward to modify Eq. (11) to encourage $v$ to point in specified directions (Hug et al., 2015):

$$\min_{\rho_t, v_t} \int_0^1 \int_\Omega v_t(x)^T A(x) v_t(x) \rho_t(x) \, dx \, dt. \tag{12}$$

Using the language developed in Appendix A, $A(x) \succ 0$ specifies a metric $g$ on the Riemannian manifold $\mathcal{M} = \left(\mathbb{R}^D, g\right)$, and $v_t(x)^T A(x) v_t(x) = \|v_t(x)\|_{A(x)}^2$. Eq. (12) encourages $v$ to be aligned with the eigenvectors $u_1$ corresponding to the minimal eigenvalues $\lambda_1$ of the matrices $A(x)$.

While works like (Hug et al., 2015; Zhang et al., 2022) investigate the modeling applications of anisotropic optimal transport, they only consider the case where the Riemannian metric $A(x)$ is available a priori. This assumption is unrealistic for many problem domains, motivating our model, which learns a metric from cross-sectional samples from populations evolving over time.

# D EXPERIMENT DETAILS

## D.1 METRIC RECOVERY

**Circular data.** The ground truth metric tensor for this example is $A(x,y) = I - v(x,y)v(x,y)^T$, where $v(x,y) = \left( \frac{-y}{\sqrt{x^2+y^2}}, \frac{x}{\sqrt{x^2+y^2}} \right)$.

To generate the training data, we begin by drawing 100 samples each from 4 isotropic normal distributions with standard deviation $\sigma = 0.1$ whose means are $\mu_i \in \{(1,0),(-1,0),(-1,-1),(0,1)\}$. We randomly pair samples from subsequent distributions and compute $A$-geodesics between each pair by solving problem (8). We implement (8) in Pytorch using a time-invariant vector field $v_\theta$ parametrized by a fully connected two-layer neural network with ELU nonlinearities and 64 hidden dimensions. We set $\lambda = 1$ and solve the initial value problem $\dot{x}(t) = v_\theta(x(t)); x(0) = x_0$ using the explicit Adams solver in torchdiffeq's odeint with default hyperparameters (Chen et al., 2018a). We optimize the objective using AdamW with learning rate $10^{-3}$ and weight decay $10^{-3}$ and train for 100 epochs per pair of samples. We then draw 24 points at equispaced times $t_i \in [0,1]$ from each resulting geodesic and aggregate across geodesics to form the observed populations $X^{t_i}$.

We then use our method to recover $\hat{A}^{-1}(x)$ from the $X^{t_i}$. We parametrize the scalar potentials in (4) as a single-hidden-layer neural net with 32 hidden dimensions and Softplus activation. We parametrize the matrix field $\hat{A}^{-1}(x)$ as $A^{-1}(x) = Q(x)^T Q(x) + 10^{-3}I$, where $Q(x)$ is a two-layer neural network with Softplus activations and 32 hidden dimensions. The strength of the gradient penalty (5) is $10^{-3}$ when training the potentials $\phi$ in the first step of alternation and $10^{-4}$ in the next step; it is 1 when training $\hat{A}^{-1}$. The strength of the regularization (6) is $10^3$. We carry out a two steps of the alternating scheme by training with AdamW with learning rate $10^{-2}$ and weight decay $5 * 10^{-1}$. We train for 300 epochs for the $\phi$ step and 1,000 epochs for the $A$ step.

We evaluate the alignment score $\ell(A, \hat{A})$ on a $100 \times 100$ grid overlaid on a box of radius 1.5.

**Mass splitting data.** The ground truth metric tensor for this example is $A(x,y) = I - v(x,y)v(x,y)^T$, where $v(x,y) = \left( \frac{1}{\sqrt{2}}, \frac{1}{\sqrt{2}} \right)$ for $y \geq 0$ and $v(x,y) = \left( \frac{1}{\sqrt{2}}, \frac{-1}{\sqrt{2}} \right)$ for $y < 0$.

To generate the training data, we begin by drawing 100 samples each from a standard normal distribution and from a mixture of two isotropic normal distributions with unit variance and mixture components centered at $(10, 10)$ and $(10, -10)$. We randomly pair samples from subsequent distributions and compute $A$-geodesics between each pair by solving problem (8) using the method described in the circular data section. We then draw 10 points at equispaced times $t_i \in [0,1]$ from each resulting geodesic and aggregate across geodesics to form the observed populations $X^{t_i}$.

We then use our method to recover $\hat{A}^{-1}(x)$ from the $X^{t_i}$. We parametrize the scalar potentials in (4) as a single-hidden-layer neural net with 32 hidden dimensions and Softplus activation. We parametrize the matrix field $\hat{A}^{-1}(x)$ as $A^{-1}(x) = Q(x)^T Q(x) + 10^{-3}I$, where $Q(x)$ is a two-layer neural network with Softplus activations and 32 hidden dimensions. The strength of the gradient penalty (5) is 2 when training the potentials $\phi$ and 1 when training $\hat{A}$. The strength of the regularization (6) is $10^6$. We carry out a single step of the alternating scheme by training with AdamW with learning rate $5 * 10^{-3}$ and weight decay $10^{-3}$. We train for 600 epochs for the $\phi$ step and 20,000 epochs for the $A$ step.

We evaluate the alignment score $\ell(A, \hat{A})$ on a $100 \times 100$ grid overlaid on the rectangular region $[-2.5, 15] \times [-15, 15]$.

**X-Paths data.** The ground truth metric tensor for this example is $A(x,y) = I - v(x,y)v(x,y)^T$. Here we define $v(x,y) = \alpha(x,y)v_1(x,y) + \beta(x,y)v_2(x,y)$, where $v_1(x,y) = \left( \frac{1}{\sqrt{2}}, \frac{1}{\sqrt{2}} \right)$ and $v_2(x,y) = \left( \frac{1}{\sqrt{2}}, \frac{-1}{\sqrt{2}} \right)$. We then define $\alpha(x,y) = 1.25 \tanh(\text{ReLU}(x \cdot y))$ and $\beta(x,y) = -1.25 \tanh(\text{ReLU}(-x \cdot y))$. Intuitively, $\alpha$ should be large in quadrants 1 and 3 and $\beta$ should be large in quadrants 2 and 4.

The training data for this example consists of two trajectories. To generate it, we begin by drawing 100 samples each from isotropic normal distributions with standard deviation $\sigma = 0.1$ centred at $(-1, -1), (1, 1)$ for the first trajectory and $(-1, 1), (1, -1)$ for the second trajectory. We randomly pair samples from subsequent distributions along each trajectory and compute $A$-geodesics between each pair by solving problem (8) using the method described in the circular data section. We then draw 10 points at equispaced times $t_i \in [0, 1]$ from each resulting geodesic and aggregate across geodesics to form the observed populations $X^{t_i}$.

We then use our method to recover $\hat{A}(x)$ from the $X^{t_i}$. We parametrize the scalar potentials in (4) as a single-hidden-layer neural net with 32 hidden dimensions and Softplus activation. We parametrize the matrix field $\hat{A}(x)$ as $A(x) = Q(x)^T Q(x) + 10^{-3} I$, where $Q(x)$ is a two-layer neural network with Softplus activations and 32 hidden dimensions. The strength of the gradient penalty (5) is $10^{-3}$ when training the potentials $\phi$ in the first alternating step and $10^{-4}$ for subsequent steps. It is 1 when training $\hat{A}$. The strength of the regularization (6) is $10^3$. We carry out three steps of the alternating scheme by training with AdamW with learning rate $10^{-2}$ and weight decay $5 * 10^{-3}$. We train for 300 epochs for the $\phi$ step and 40,000 epochs for the $A$ step.

We evaluate the alignment score $\ell(A, \hat{A})$ on a $100 \times 100$ grid overlaid on a box of radius 1.5.

## D.2   SCRNA TRAJECTORY INFERENCE

**Data pre-processing.** The data for this experiment consists of force-directed layout embedding coordinates of gene expression data from Schiebinger et al. (2019) collected over 18 days of reprogramming (39 time points total) which we rescale by a factor of $10^{-3}$ for increased stability in training. We construct populations $X^{t_i}$ for $i = 1, ..., 39$ by randomly drawing 500 samples per time point in the original data; this corresponds to using 8.25% of the available data on average.

**Learning the metric.** We use our method to learn a metric $\hat{A}^{-1}(x)$ from the $X^{t_i}$. We parametrize the scalar potentials in (4) as a single-hidden-layer neural net with 128 hidden dimensions and Softplus activation. We parametrize the matrix field $\hat{A}^{-1}(x)$ as $A^{-1}(x) = Q(x)^T Q(x) + 10^{-9} I$, where $Q(x)$ is a single-hidden-layer neural network with Softplus activation and 2048 hidden dimensions. We omit the gradient penalty (5) when training the potentials $\phi$ and set the penalty strength to 10 when training $\hat{A}$. The strength of the regularization (6) is $5 * 10^2$. We carry out a single step of the alternating scheme by training with AdamW with learning rate $5 * 10^{-3}$ and weight decay $1.5 * 10^{-2}$. We train for 100 epochs for the $\phi$ step and 5,000 epochs for the $A$ step.

**Trajectory inference task.** We parametrize the time-varying velocity field $v_{t, \theta}$ as a fully connected three-hidden-layer neural network with 64 hidden dimensions and Softplus activations. We follow Grathwohl et al. (2019) and concatenate the time variable to the input to each layer of the neural net. We compute particle trajectories $x(t)$ by solving the initial value problem $\dot{x}(t) = v_\theta(x(t)); x(0) = x_0$ for initial positions $x_0 \in X^{t_i}$ using the midpoint method solver in torchdiffeq's `odeint` with default hyperparameters (Chen et al., 2018a). The fitting loss for this task is GeomLoss's Sinkhorn divergence with $p = 2$ and blur parameter fixed to $5 * 10^{-2}$. We fix $\lambda = 10^{-1}$ for both our learned metric $\hat{A}$ and for the identity baseline $A = I$ and choose the intermediate time points $t_j$ to be an equispaced sampling of $[0, 1]$ with step size $\frac{1}{60}$ for all experiments.

We solve problem (7) for each pair of subsequent time retained points $(t_i, t_{i+1})$. In each case, we optimize the objective using AdamW with learning rate $10^{-3}$ and weight decay $10^{-3}$ and train for 10,000 iterations. We evaluate the inferred trajectories by approximately computing the $W_1$ distance (using GeomLoss with $p = 1$ and blur of $10^{-6}$) between left-out time points in the ground truth data and advected samples at corresponding time points. For left-out time points of form $t = ki + \ell$ for some integer $\ell \in 1, ..., k - 1$, the corresponding advected sample in the equispaced sampling of step size $\frac{1}{60}$ of $[0, 1]$ has index $\lfloor \frac{j}{k} * 60 \rfloor$.

## D.3   SNOW GOOSE TRAJECTORY INFERENCE

**Data pre-processing.** The training data for learning the metric in this experiment consists of time-stamped sightings of untagged snow geese (*Anser caerulescens*) across the U.S. and Canada during their spring migration. This data is drawn from the February 2022 version of the eBird basic

dataset (Sullivan et al., 2009). We bin the sightings by month of observation, keep 1000 records per month, and convert the sighting locations from latitude-longitude to $xy$ coordinates using the Mercator projection implemented in Matplotlib's Basemap module with `rsphere` set to 5 and the latitude/longitude of the lower left and upper right corners of the projection set to the minimum latitude and longitude and maximum latitude and longitude in the training data, respectively.

The ground truth goose trajectories are drawn from the "Banks Island SNGO" study on Movebank. This data consists of time-stamped GPS measurements of the locations of 8 snow geese from 2019 to 2022. We use the same Mercator projection to convert the goose location measurements from latitude/longitude to $xy$ coordinates. We then estimate the initial and final time point of a single migration for each goose. For geese with ID $\{82901, 82902, 82905, 82906, 82907, 82908, 82909, 82910\}$, their respective initial time indices are $\{20000, 0, 0, 20000, 0, 0, 0, 0\}$ and their respective final time indices are $\{26000, 9100, 15057, 26500, 9000, 13037, 7201, 10000\}$. The initial location of each goose is the initial condition $x_0$ in (8), and the final location of each goose is $x_1$.

**Learning the metric.** We use our method to learn a metric $\hat{A}^{-1}(x)$ from the $X^{t_i}$. We parametrize the scalar potentials in (4) as a single-hidden-layer neural net with 32 hidden dimensions and a Softplus activation. We parametrize the matrix field $\hat{A}^{-1}(x)$ as $A^{-1}(x) = Q(x)^T Q(x) + 10^{-3}I$, where $Q(x)$ is a two-layer neural network with Softplus activations and 32 hidden dimensions. The strength of the gradient penalty (5) is $10^{-6}$ when training the potentials $\phi$ and 1 when training $\hat{A}^{-1}$. The strength of the regularization (6) is $10^9$. We carry out a single a step of the alternating scheme by training with AdamW with learning rate $10^{-2}$ and weight decay $10^{-3}$. We train for 2,000 epochs for the $\phi$ step and 10,000 epochs for the $A$ step.

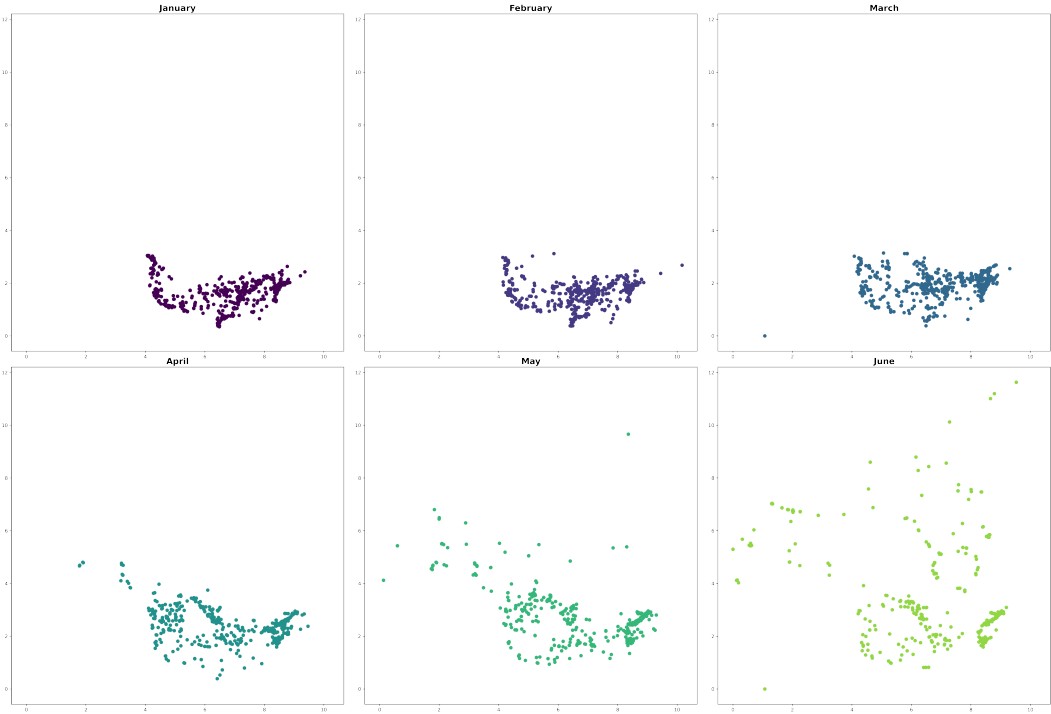

Figure 5: The untagged goose sighting data used to learn the metric for the bird trajectory inference experiments. Each subplot depicts goose sightings in the U.S. and Canada during one month of the spring migration, beginning in January (upper-left) and ending in June (bottom-right). Note that there is no correspondence between points in subsequent time points; any given goose is likely to have been sighted only once.

**Trajectory inference task.** We parametrize the time-varying velocity field $v_{t,\theta}$ as a fully connected three-hidden-layer neural network with 64 hidden dimensions and ELU nonlinearities. We compute particle trajectories $x(t)$ by solving the initial value problem $\dot{x}(t) = v_\theta(x(t)); x(0) = x_0$ for initial positions $x_0 \in X^{t_i}$ using the dopri5 solver in torchdiffeq's `odeint` with default hyperparameters

(Chen et al., 2018a). We fix $\lambda = 1.25 * 10^2$ for geese 82901, 82902, 82906, 82908, and 82909 and use $\lambda = 2.5 * 10^2$ for the remaining geese 82905, 82907, 82910. In each case, we optimize the objective using AdamW with learning rate $10^{-3}$ and no weight decay, set the times $t_j$ in (8) be an equispaced sampling of $[0, 1]$ with 32 time points and train for 500 iterations.

## E   ABLATION STUDY: IMPACT OF THE REGULARIZATION COEFFICIENT

In this section, we carry out an ablation study of the impact of the coefficient $\lambda$ on the regularization term $R(A)$ in (4). We repeat the metric recovery experiment on the "Circular" example. We follow the procedure in Appendix D.1 but set $\lambda \in \{5 * 10^2, 10^3, 5 * 10^3, 10^4, 5 * 10^4, 10^5\}$. (Note that our reported results in Section 4.1 use $\lambda = 10^3$.)

The results of this experiment are recorded in Figure 6. The eigenvectors of the learned metric $\hat{A}(x)$ are robust to the value of $\lambda$ except at the largest value of $\lambda = 10^5$, where the alignment score falls to 0.857. As expected, the log-condition number of $\hat{A}(x)$ increases somewhat with $\lambda$, indicating that larger values of $\lambda$ favor high levels of anisotropy.

## F   ABLATION STUDY: REMOVING THE OT TERM

In this section, we briefly demonstrate that if we remove the optimal transport term from (4), the resulting metric is not informed by the training data. We repeat the metric recovery experiment on the "Circular" example. We follow the procedure in Appendix D.1 but exclude the OT term

$$\frac{1}{K} \sum_{k=1}^{K} \left( \int_{\mathcal{M}} \phi^k(x) d\rho_0^k(x) - \int_{\mathcal{M}} \phi^k(x) d\rho_1^k(x) \right) \tag{13}$$

from our implementation of (4). The results of this experiment are presented in Figure 7. Note that the learned metric is no longer informed by the data; the low-energy eigenvectors point in the direction of the vertical axis everywhere and the log-condition number is simply an increasing function of distance from the origin.

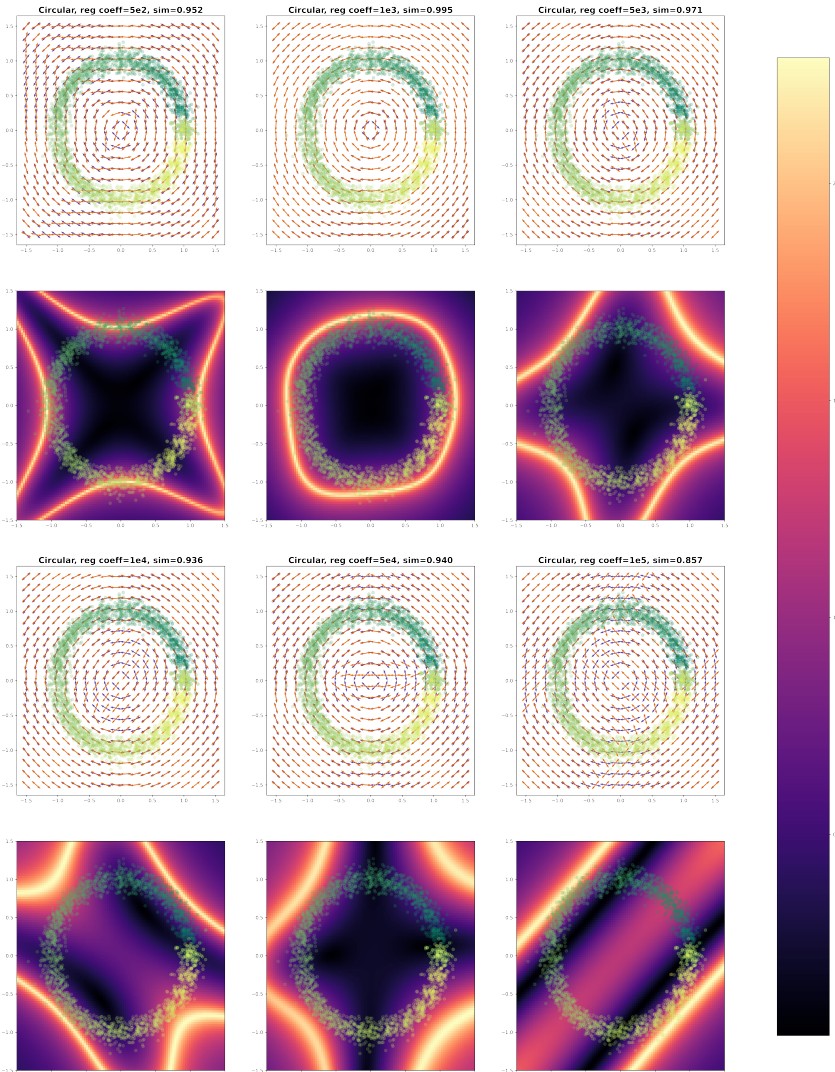

Figure 6: Results of ablation study of the impact of regularization coefficient $\lambda$ in (4). The eigenvectors of the learned metric $\hat{A}(x)$ are robust to the value of $\lambda$ (rows 1,3). The log-condition number of $\hat{A}(x)$ increases with $\lambda$ (rows 2,4), indicating that large values of $\lambda$ lead to increased anisotropy.

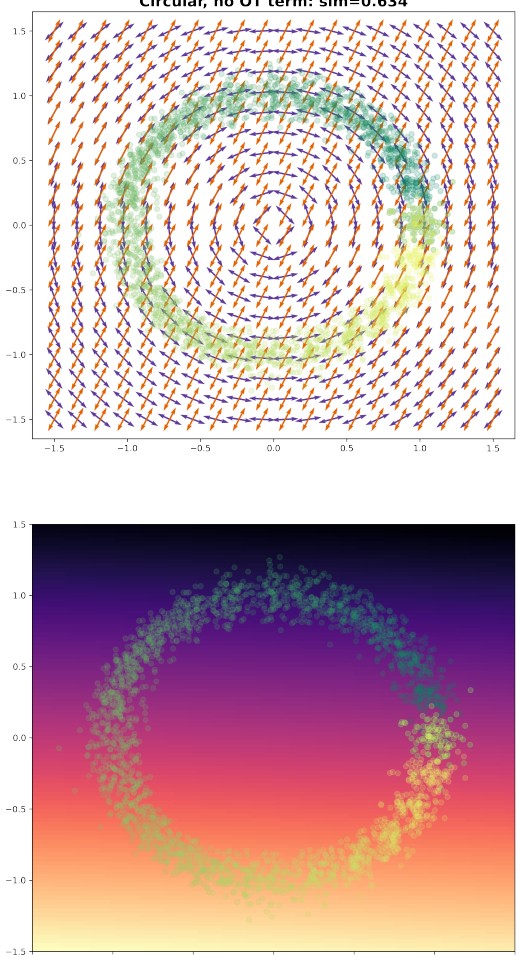

Figure 7: Learned metric $\hat{A}(x)$ with the OT term (13) removed from Problem (4). The metric is no longer informed by the data when the OT term is excluded.

