# OpenReview forum: "Riemannian Metric Learning via Optimal Transport"
_ICLR.cc/2023/Conference — ICLR 2023 poster_

### Official Review · Reviewer_QfVV · 2022-10-24

**Confidence:** 2
**Correctness:** 3
**Technical Novelty And Significance:** 4
**Empirical Novelty And Significance:** 2
**Recommendation:** 5

**Clarity, Quality, Novelty And Reproducibility:**

I found the paper difficult to read as I missed explanations on how to go from one equation to another. Perhaps readers with a stronger background in optimal transport will not have such issues, but I had trouble grasping the foundation of the work.

The result, however, seems really neat and quite novel.

**Strength And Weaknesses:**

I found the paper difficult to read and have low confidence on the strengths and weaknesses listed below.

## Strengths
* The key technical innovation appears to be a minimax objective (5) for Riemannian metric learning that avoids the need for computing geodesics under the current estimate of the metric. This is potentially a big deal as computing geodesics is both expensive and unstable. I did however not understand the derivation of this objective, so I may be missing something here.

## Weaknesses
* I found the paper difficult to read as I found it non-trivial how to get from one equation to another. I get that sometimes these steps are indeed non-trivial, but a bit of hand-holding would be nice. Specifically, I did not understand how to go from Eq 3 to Eq 4 and from there to Eq 5.
* As far as I could tell, the regularizer in Eq. 7 is going to be expensive to evaluate as I suppose you have to numerically invert the metric (cubic complexity). Is this a correct reading?
* I missed citations to some of the many papers on learning Riemannian metrics over $\mathbb{R}^D$. Even if the applied strategies are different than the current paper, I still think previous work should be acknowledged. See for example "Learning Riemannian Metrics", Lebanon (UAI 2003) or "A Locally Adaptive Normal Distribution", Arvanitidis et al., NeurIPS 2016.

## Minor comments
* I would suggest moving the related work section to a later part of the paper as I found that it took too long before I got to the actual contribution of the paper.

**Summary Of The Paper:**

The paper proposes a method for learning a Riemannian metric for trajectory data over $\mathbb{R}^D$ using optimal transport. The key technical novelty is the development of a minimax objective that avoids computing geodesics during training, thereby both improving training speed and stability.

**Summary Of The Review:**

A potentially really neat result, but I found the paper too difficult to read for me to provide a useful verdict. I do not recommend acceptance in its current form due to the difficulties I had reading the paper, but I also claim low confidence.

---

> ### Author Response · Authors · 2022-11-11
> **Response to Official Review of Paper951 by Reviewer QfVV**
>
> We thank the reviewer for their comments and are glad that they appreciate the novelty and interest of our minimax approach to Riemannian metric learning. We have uploaded a revision of our paper to OpenReview, where they will find that we have addressed many comments. We outline these changes and respond to the reviewer’s comments below.
>
> *“I found the paper difficult to read as I found it non-trivial how to get from one equation to another. I get that sometimes these steps are indeed non-trivial, but a bit of hand-holding would be nice. Specifically, I did not understand how to go from Eq 3 to Eq 4 and from there to Eq 5.”*
>
> We have updated the description of our method in Section 3.1 (previously Section 4.1) to clarify our idealized objective (1) (previously Eq 3) and the steps involved in passing from Equation 1 to Equation 2 and then to 3 (previously 3,4,5, respectively). We would be happy to provide further clarifications upon request.
>
> *“As far as I could tell, the regularizer in Eq. 7 is going to be expensive to evaluate as I suppose you have to numerically invert the metric (cubic complexity). Is this a correct reading?”*
>
> As the metric $A$ only appears in our optimization problem via its inverse, we directly parametrize $A^{-1}$ via a neural network in training and only invert the learned $A^{-1}$ where we require the evaluation of $A$ in downstream tasks. As such, we avoid the need to invert the metric during training. We have updated the subsection “Optimization scheme” of Section 3.2 (previously Section 4.2) to clarify this point.
>
> *“I missed citations to some of the many papers on learning Riemannian metrics over R^D. Even if the applied strategies are different than the current paper, I still think previous work should be acknowledged. See for example "Learning Riemannian Metrics", Lebanon (UAI 2003) or "A Locally Adaptive Normal Distribution", Arvanitidis et al., NeurIPS 2016.”*
>
> We have added a new subsection titled “Riemannian metric learning” to Section 2 (“Related Work”) that includes references to these existing works.
>
> **Conclusion.**
>
> We thank the reviewer for their comments and hope that we have adequately addressed their concerns. If they are satisfied with our answers and our revision, we respectfully request that they raise their score for this paper. Otherwise, we would be pleased to continue this discussion during the reviewer-author discussion period.

---

> ### Author Response · Authors · 2022-11-16
> **Reminder for Reviewer QfVV**
>
> Dear Reviewer QfVV -- as the first stage of the discussion period will close in a few days, we would greatly appreciate if you'd take a look at our response to your review soon and let us know if you would like to see further changes to our manuscript. We look forward to addressing your remaining concerns before the end of this stage of discussion. If our response was satisfactory, we ask that you consider raising your score for our submission. Thank you for your time.

---

### Official Review · Reviewer_m2sP · 2022-10-25

**Confidence:** 2
**Correctness:** 3
**Technical Novelty And Significance:** 3
**Empirical Novelty And Significance:** 2
**Recommendation:** 6

**Clarity, Quality, Novelty And Reproducibility:**


The paper is difficult to follow. I have tried to formulate the basic optimization problem addressed here through multiple readings and I am still not sure if I understood the idea. There has to be simpler, more direct way of explaining the problem to the audience before getting into other details.

**Strength And Weaknesses:**

Strengths:

Learning Riemannian metrics from data is an important and commonly studied problem. The use of a time series data for metric learning makes this paper more interesting.

Weakness:

The paper makes for a confusing read. Perhaps it is intended for an audience where the basic setup is already well known.

(1)	What is the justification behind assuming that the samples follow geodesic paths under some Riemannian metric?
(2)	Why solve for the parallel transport problem rather than any other metric (e.g. Fisher-Rao metric) on the space of distribution space?


**Summary Of The Paper:**

This is a relatively difficult paper to follow. Despite several attempts I have more questions about the paper than answers.  My review is based on my limited understanding of the ideas presented here. The goal here seems to be to analyze data (samples) observed from temporal evolution of a probability distribution. The assumption is that individual samples evolve in time along geodesic paths (with respect to certain nonstandard metric). Thus, the paper seeks to estimate this Riemannian metric from these temporal evolutions. However, one does not have the temporal registration between samples across times, i.e., one does not know which sample at time t matches with which sample at time t+1. Thus, the paper casts it as problem of comparing distributions (approximated by empirical distributions) and hence the use of Monge’s transport problem.  There is some related discussion in the paper on spatial regularization of the metric tensor field. The actual optimization over the metric tensor field is performed using neural network. The paper is motivated by applications in molecular biology, esp. in analyzing scRNA data where there is past literature on using optimal transport for estimating trajectories.

**Summary Of The Review:**


This is a paper about learning Riemannian metric using time series data. The paper is difficult to read and understand. Given my limited understanding, it is using the Monge transport problem for solving for the unknown metric tensor field. The paper presentation and justification of the approach needs improvement.

---

> ### Author Response · Authors · 2022-11-11
> **Response to Official Review of Paper951 by Reviewer m2sP**
>
> We thank the reviewer for their time and are glad that they share our interest in learning Riemannian metrics from time series data. We have uploaded a revision of our paper to OpenReview, where they will find that we have addressed many comments. We outline these changes and respond to the reviewer’s comments below.
>
> *“What is the justification behind assuming that the samples follow geodesic paths under some Riemannian metric?”*
>
> In our problem, we assume that we only observe the initial and final spatial distributions of populations evolving on the manifold whose metric we seek to learn. Since we do not observe complete trajectories, we must make some assumption on the structure of the paths taken by samples between their initial and final positions. Geodesics are paths that minimize the action (or average kinetic energy) of a particle traveling between a pair of points $(x,y)$ on a manifold; this least-action interpretation of a geodesic makes it a natural prior on paths in the absence of further information. We have updated Section 3.1 (previously Section 4.1) to clarify this point.
>
> *“Why solve for the parallel transport problem rather than any other metric (e.g. Fisher-Rao metric) on the space of distribution space?”*
>
> In our model, we do not know the continuous dynamics that samples follow. Consequently, we do not not know the maps $r^k$ in Equation (1) that specify the correspondence between particle positions $x$ at $t=0$ and positions $r^k(x)$ at final times $T_k$. Furthermore, particles that we observe at $t=0$ may no longer be in the sample at $t=T_k$. This issue is unavoidable for destructive measurement processes such as scRNA sequencing, where cells whose scRNA profile is observed at $t=0$ would be destroyed at this time and hence unobservable at a future time $t=T_k$.
>
> To accommodate these limitations, we replace the true matchings of initial and final positions $r^k$ with the Monge map, which matches units of mass from the initial and final distributions $\rho^k_0, \rho^k_1$ to minimize their average $A$-geodesic distance. As the Monge map is the solution to an optimal transport problem, the inner problem in our resulting objective has the structure of an optimal transport problem. We have updated Section 3.1 (previously Section 4.1) in our revised manuscript to further clarify these points.
>
> **Conclusion.**
>
> We thank the reviewer for their comments and hope that we have adequately addressed their concerns. If they are satisfied with our answers and our revision, we respectfully request that they raise their score for this paper. Otherwise, we would be pleased to continue this discussion during the reviewer-author discussion period.

---

> ### Author Response · Authors · 2022-11-16
> **Reminder for Reviewer m2sP**
>
> Dear Reviewer m2sP -- as the first stage of the discussion period will close in a few days, we would greatly appreciate if you'd take a look at our response to your review soon and let us know if you would like to see further changes to our manuscript. We look forward to addressing your remaining concerns before the end of this stage of discussion. If our response was satisfactory, we ask that you consider raising your score for our submission. Thank you for your time.

---

### Official Review · Reviewer_v7rg · 2022-10-28

**Confidence:** 3
**Clarity, Quality, Novelty And Reproducibility:** Reasonable quality, poor clarity very…
**Correctness:** 3
**Technical Novelty And Significance:** 3
**Empirical Novelty And Significance:** 3
**Recommendation:** 5

**Strength And Weaknesses:**

Strengths

- I find the problem setup and the solution formulation to be very interesting and refreshing. Changing the metric and having to learn that from sparse trajectory data is innovative!
- The results do showcase a proof-of-concept, as most claims made about improvements with a learned metric instead of a trivial one are indeed so in the challenging problems of the experiments (especially Figures 1,2 and 3)

Weaknesses

- Overall the paper lacks global quality in terms of clarity of discussion. The parts corresponding to the scRNA problem are poorly introduced and could do with a better background.
- Occasionally the discussion feels very hurried in the paper (especially in section 5.9). Also, how do Schiebinger et al. (2019) and Tong et al. (2020) compare with the result in Figures 2 and 3?
- In addition, the choice of regularizer and the need to parameterize the metric and function \phi using neural networks feels a bit arbitrary to me. What is the motivation here? in addition to easy use of gradient-based optimization?
- The result in Figure 4 is not convincing. Ideally, I suppose the predicted trajectories must resemble the ground truth very closely.  but they clearly are a bit too smooth and don't have the high frequencies like the ground truth. Could there be a baseline comparison to show the proposed improvements?
- Some minor issues - all figures could do with bigger sizes (especially Figure 2) and with much bigger font size for the titles.

**Summary Of The Paper:**

This paper proposes to solve a dynamical optimal transport problem where: given a sequence of probability measures sampled in time, the goal is to recover an accurate trajectory of the measures for all intermediate times. Such a scenario is highlighted over two practical applications - single-cell RNA sequencing and discovering migratory paths of birds based on sparse information about their intermediate geographic locations.

The paper proposes to model these measures over a general Riemannian manifold instead of a Euclidean space where each point is now endowed with a metric tensor. The eventual goal is then to *learn* this metric from samples and demonstrate improved trajectory prediction. The authors then formulate the trajectory inference problem as an optimal transport problem over the manifold and make use of the dual formulation to find the most convenient way to learn the metric tensor field. Additionally, they also propose a regularizer that is claimed to aid the learning process.

Experiments are shown for 1 synthetic and 2 actual datasets - scRNA and bird migration. The results show improvement over simple baselines like trajectory inference using the Euclidean metric or using no regularization


**Summary Of The Review:**

Overall, I think this paper is below the threshold for acceptance. Despite rating highly on problem formulation and some clever tricks on using dual problems for convenient estimation of the metric - (1.) There is an occasional lack of background and the discussion lacks structure (2.) More importantly, the results are not super convincing beyond a simple proof of concept. Like specifically - the mass splitting example in Figure 1, and results for bird migration in Figure 4, the lack of baselines in Figure 2, etc.

---

> ### Author Response · Authors · 2022-11-11
> **Response to Official Review of Paper951 by Reviewer v7rg (part 1/2)**
>
> We thank the reviewer for their insightful review and for their suggestions, which will improve the quality of our manuscript. We are also glad that they appreciate the innovativeness of our problem setup and solution formulation.
>
> We have uploaded a revision of our paper to OpenReview, where they will find that we have addressed many comments. We outline these changes and respond to the reviewer’s comments below.
>
> *“The parts corresponding to the scRNA problem are poorly introduced and could do with a better background.”*
>
> We have expanded the discussion of the scRNA trajectory inference problem in Section 4.2 of the updated manuscript (previously Section 5.2). We hope that this sufficiently clarifies the problem and our contributions.
>
> *“Also, how do Schiebinger et al. (2019) and Tong et al. (2020) compare with the result in Figures 2 and 3?”*
>
> We have added a baseline comparison for scRNA trajectory inference to Section 4.2 (Section 5.2 in the previous version of the manuscript) that is inspired by the approach of Schiebinger et al. As their Waddington OT method solves a static optimal transport problem and pushes cells through transport maps between fixed time steps, a direct comparison against our dynamical method is not possible. However, we aim to replicate their approach as closely as possible and show that their method results in inferred trajectories of worse quality than our approach that leverages a learned metric; we provide details in Section 4.2 of our updated manuscript.
>
> Our “Without $A$” and “$A$ = Id” baselines were obtained from a model that is comparable to the “Base” and “Base + E” models, respectively, from Tong et al. Both our baseline model and their models learn a velocity field that pushes samples from a base distribution (data at some time $t_\ell$) onto samples from a target distribution (data at a future time $t_{\ell + 1}$) and treat the path followed by these samples as they flow through the velocity field as an interpolation between the base and target distributions. Our “Without $A$” baseline and their “Base” model do not regularize the velocity field at all, whereas our “$A$ = Id” baseline and their “Base + E” model regularize the velocity field by penalizing its squared norm, which encourages samples to flow along straight paths. We have updated our discussion in Section 4.2 to clarify these points.
>
> The primary difference between our baseline and Tong et al. 's models is that we use the Sinkhorn divergence as a fitting loss to encourage the base samples to reach their targets, whereas Tong et al. use log-likelihood as a fitting loss. We do this to ensure a fair comparison with our own approach, which uses a Sinkhorn divergence as a fitting loss (see Eq. (7) in our manuscript). We opt for the Sinkhorn divergence in our approach because we found that it leads to stabler training than a log-likelihood fitting loss.
>
> *“In addition, the choice of regularizer and the need to parameterize the metric and function $\phi$ using neural networks feels a bit arbitrary to me. What is the motivation here? in addition to easy use of gradient-based optimization?”*
>
> As noted in subsection “Choice of regularizer $R(A)$” of Section 3.2 in the updated manuscript, we seek a regularizer that prevents the objective in Equation (4) from being driven to 0 by choosing a metric $A(x) = \alpha I$ for $\alpha \rightarrow 0$. The trivial solution we avoid would incorporate no useful information from the observed samples from the distributions $\rho^k_0$ and $\rho^k_1$, as it is a rescaling of the standard Euclidean metric. Penalizing $\|A^-1\|^2_F$ is a natural way to exclude such solutions, since $\|A^-1\|^2_F \rightarrow \infty$ as $A \rightarrow 0$.
>
> If the reviewer has another regularizer in mind, we would be happy to investigate its effectiveness in our setting. We have updated our discussion in this subsection to clarify these points.
>
> We indeed parametrize the metric and the potentials $\phi^k$ using neural networks to make our problem tractable via gradient-based optimization. Neural parametrizations are a standard technique in applied optimal transport, and as noted in subsection “Enforcing the Lipschitz constraint” in Section 3.2 of the updated manuscript, Korotin et al. have shown that neural parametrizations of Kantorovich potentials with our Lipschitz penalty (Equation (5)) provide good approximations to the directions of the gradients of the true Kantorovich potentials. This is sufficient for our purposes, as these gradients are the key input to the outer optimization problem and determine the low-energy eigenvectors of the learned metric $A(x)$.
>
> **We continue our response in part 2/2.**

---

> ### Author Response · Authors · 2022-11-11
> **Response to Official Review of Paper951 by Reviewer v7rg (part 2/2)**
>
> **This is part 2/2 of our response.**
>
> *“The result in Figure 4 is not convincing. Ideally, I suppose the predicted trajectories must resemble the ground truth very closely. but they clearly are a bit too smooth and don't have the high frequencies like the ground truth. Could there be a baseline comparison to show the proposed improvements?”*
>
> We have updated Figure 4 to include a visualization of the straight-line trajectories that constitute our baseline comparison; these correspond to geodesics under a Euclidean metric. As we note in subsection “Snow goose data” of Section 4.3 (previously Section 5.3), our learned metric results in a 26.9% reduction in dynamic time warping distance between inferred and ground truth trajectories relative to the straight-line baseline.
>
> We have also included a figure in the Appendix D.3 (Figure 5) depicting the data from which we learn our metric for the snow goose experiments. Note that there is no correspondence between goose observations at subsequent time points; we learn our metric using exclusively a time series of six highly diffuse point clouds representing untagged goose sightings in each month of the spring migration.
>
> We would not expect such observations to contain enough information to predict high-frequency detail in the goose trajectories, but this data is far cheaper to obtain than complete migratory trajectories, which are typically recorded via GPS trackers attached to individual geese. The widespread availability and low cost of obtaining untagged bird sighting data motivates the use of our method for bird trajectory inference. We have updated Section 4.3 to clarify these points.
>
> As noted in the introduction to Section 4.3, the learned metric summarizes the factors that birds may use to modify their migratory paths locally, such as local weather conditions, food availability, and predatory pressures. The path of any one goose is of course impossible to predict at the micro scale given only the endpoints of their travel over several months, so we do not expect to reproduce the ground truth trajectories of individual geese with high fidelity.
>
> **Conclusion.**
>
> We thank the reviewer for their comments and hope that we have adequately addressed their concerns. If they are satisfied with our answers and our revision, we respectfully request that they raise their score for this paper. Otherwise, we would be pleased to continue this discussion during the reviewer-author discussion period.

---

> ### Author Response · Authors · 2022-11-16
> **Reminder for Reviewer v7rg**
>
> Dear Reviewer v7rg -- as the first stage of the discussion period will close in a few days, we would greatly appreciate if you'd take a look at our response to your review soon and let us know if you would like to see further changes to our manuscript. We look forward to addressing your remaining concerns before the end of this stage of discussion. If our response was satisfactory, we ask that you consider raising your score for our submission. Thank you for your time.

---

### Official Review · Reviewer_M27J · 2022-11-03

**Confidence:** 2
**Correctness:** 4
**Technical Novelty And Significance:** 4
**Empirical Novelty And Significance:** 4
**Recommendation:** 8

**Clarity, Quality, Novelty And Reproducibility:**

It is clearly written but it may be a bit abstract sometimes for readers not very familiar with the literature. I also believe the work is novel and reproducible.

**Strength And Weaknesses:**

I believe the paper proposes an interesting use-case of learning the metric. As such it is theoretically sound. I would like to mention that it is not in my expertise to dissect and digest the results completely. The optimization problem is still abstract. It would be great to discuss it a few more details about the precise optimization steps and updates strategies.


**Summary Of The Paper:**

The paper deals with identifying a Riemannian metric for the cross-sectional samples of evolving probability measures on a common manifold. In particular, it makes use of  optimal transport theory to compute the Riemannian metric (that minimizes the 1-Wasserstein distance). The usefulness of the learned metric is shown in different applications.


**Summary Of The Review:**

I wish to point out that the paper is not in my domain of expertise. I, however, believe that the work is a novel contribution to and should be interesting to the community.

---

> ### Author Response · Authors · 2022-11-11
> **Response to Official Review of Paper951 by Reviewer M27J**
>
> We thank the reviewer for their comments and are glad that they appreciate the novelty and interest of our method. We have uploaded a revision of our paper to OpenReview, where we have added additional details to Sections 3.1 and 3.2 that further clarify our model and optimization strategy, respectively. We hope that this is helpful to the reviewer and are happy to provide additional clarifications upon request.

---

### Decision · Program_Chairs · 2023-01-20

**Decision:**

Accept: poster

**Justification For Why Not Higher Score:**

The paper is interesting and  provide novel contributions but the novelty is not groundbreaking and will interest only a subpart of the ML community

**Justification For Why Not Lower Score:**

The paper provide interesting  novel contributions.

**Metareview: Summary, Strengths And Weaknesses:**

The paper addresses the problem of learning the riemannian metric of evolving distributions when the evolution is characterized by a Monge Map. The authors succeed in formalizing an interesting  learning problem (including new regularizers and approximation of constraint) and  in proposing a  numerical algorithm solving it. After discussions, we believe that the paper addresses an interesting problem
with a relevant application and deserves to be presented at ICLR. We also urge the authors to improve clarity of the final version of the paper.

**Note From Pc:**

if the above contains the word "oral" or "spotlight" please see: "oral" presentation means -> notable-top-5% and "spotlight" means -> notable-top-25%. As stated in our emails, we are disassociating presentation type from AC recommendations